



# The Pan-Arctic Catchment Database (ARCADE)

Niek Jesse Speetjens[1], Gustaf Hugelius[2], Thomas Gumbricht[2], Hugues Lantuit[3], Wouter R. Berghuijs[1], Philip A. Pika[1], Amanda Poste[4], Jorien E. Vonk[1]

[1]Vrije Universiteit Amsterdam (VUA), Department of Earth Sciences, Earth and Climate Cluster,
Amsterdam, 1081 HV Amsterdam, The Netherlands
[2]Stockholm University (SU), Department of Physical Geography,106 91 Stockholm, Sweden
[3]Alfred Wegener Institute (AWI) Helmholtz Centre for Polar and Marine Research, Ecological
Chemistry Research Unit, 27570 Bremerhaven, Germany
[4] Norwegian Institute for Water Research (NIVA), Section for Nature based solutions and aquatic
ecology, Økernveien 94, 0579 Oslo, Norway

Correspondence to: Niek Jesse Speetjens (n.j.speetjens@vu.nl/niek.j.speetjens@gmail.com)

## Abstract

The Arctic is rapidly changing. Outside the Arctic, large-sample catchment databases have transformed
catchment science from focusing on local case studies to more systematic studies of watershed
functioning. Here we present an integrated pan-ARctic CAtchments summary DatabasE (ARCADE) of
>40,000 catchments that drain into the Arctic Ocean and range in size from 1 km² to 3.1 x 10⁶ km²
(Speetjens et al., 2022). These watersheds, delineated at a 90-m resolution, are provided with 103
geospatial, environmental, climatic, and physiographic catchment properties. ARCADE is the first
aggregated database of pan-Arctic river catchments that also includes numerous small watersheds at a
high resolution. These small catchments are experiencing the greatest climatic warming while also storing
large quantities of soil carbon in landscapes that are especially prone to degradation of permafrost (i.e.,
ice-wedge polygon terrain) and associated hydrological regime shifts. ARCADE is a key step toward
monitoring the pan-Arctic across scales and is publicly available:
https://dataverse.nl/dataset.xhtml?persistentId=doi:10.34894/U9HSPV.

## 1 Introduction

Earth's rapidly changing climate is particularly evident in the Arctic. Decreasing sea ice extent has
amplified Arctic warming, which has led to an increase in mean land-surface air temperature of 3.1° C
(three times the global average of ~1° C) over the period 1979 – 2019 (AMAP, 2021; GISTEMP Team,
2021). Under all IPCC climate scenarios, the Arctic will be substantially different by mid-century (e.g.,
less snow and sea ice, degraded permafrost, and altered ecosystems) (Overland et al., 2019). The Arctic
is important in regulating the global climate system (IPCC, 2019; IPCC, 2021) and global biogeochemical



cycles (Parmentier et al., 2017). Ongoing changes in the Arctic and their consequential impacts are both local (e.g., ecosystem changes, changing food web interactions, and potential loss of biodiversity) (Vincent, 2019) and global (e.g., changing atmospheric circulation, ocean acidification, and an altered carbon cycle) (Box et al., 2019; Yamanouchi & Takata, 2020) which raises the urgency to understand this intricate system better.

In the Arctic, marine and terrestrial systems are tightly coupled. More than 10% of global river discharge flows into the Arctic Ocean (AO), which only contains about 1% of the global ocean volume (Aagaard and Carmack, 1989; McClelland et al., 2012). In addition, river discharge transports sediment, (organic) carbon, nutrients, and contaminants (Terhaar et al., 2021) into the AO. Arctic rivers integrate over local to regional scales and are therefore useful for studying the impacts of environmental and climatic change at various scales (Holmes et al., 2012).

Permanently frozen soils (permafrost) that are rich in organic carbon (OC) (Hugelius et al., 2014; Mishra et al., 2021) underlie about 60 - 80% of the AO watershed (Zhang et al., 2000; Obu et al., 2019). Permafrost conditions have long stabilized the subsurface, but ground temperatures are now warming across the northern hemisphere (Biskaborn et al., 2019). Permafrost degradation occurs slowly through deepening of the active layer (the layer that thaws during summer and refreezes during winter) (Ran et al., 2022) or more quickly through abrupt thaw of permafrost with high ground ice contents. Both types of thaw expose soil OC to degradation, which transforms it into greenhouse gases. Thus, the thawing of permafrost can accelerate global warming but also impacts hydrological, biogeochemical, and ecological processes in Arctic ecosystems, with complex consequences for lateral transport of terrestrial material to downstream freshwater and marine systems (Vonk & Gustafsson, 2013).

Investigations of Arctic change (e.g., Schuur et al., 2015; Walvoord & Kurylyk, 2016; Liljedahl et al., 2016; Lafrenière & Lamoureux, 2019; Bruhwiler et al, 2021) critically rely on data. The "Arctic Great Rivers Observatory" initiative, that runs since 2003, is a unique data set covering the six largest Arctic rivers (McClelland et al., 2008, www.arcticgreatrivers.org). While data on these large river systems can provide important insights into Arctic change (e.g., Wild et al., 2019; Terhaar et al., 2021; Behnke et al., 2021), they do not reveal the changes that occur at finer scales. Revealing such insights requires data from smaller Pan-Arctic watersheds.

Small and medium-sized watersheds drain roughly a third of the circumpolar landmass (Holmes et al., 2012). In contrast to the watersheds of the six largest Arctic rivers (Ob', Yenisey, Lena, Kolyma, Mackenzie, Yukon), the smaller watersheds are almost exclusively underlain by continuous permafrost (Holmes et al., 2012) and are often directly located at the coast. This makes these small watersheds fundamentally different from 'The Big Six' because large rivers drain to a few coastal locations (Mann et al., 2022), while the cumulative inputs of small watersheds are spread over a much larger coastal area. In addition, given their size and proximity to the AO, the changes in these watersheds could be more rapidly transferred and substantial to the Arctic coastal ecosystem.



Outside of the Arctic, the emergence of large-sample catchment databases (e.g., Hartmann et al., 2014; Newman et al., 2015; and Alvarez-Garreton et al., 2018), which combine data from many watersheds, have transformed the field from an emphasis on local case-studies towards more systematic insights into drivers of watershed functioning. For example, large-sample watershed studies allow to reveal regional differences (and similarities) in hydrological response, make space-for-time transformations, and systematically test hypotheses. This has proven critical in, for example, understanding impacts of climate change (e.g., Berghuijs et al., 2014) and testing modeling implications (e.g., Knoben et al., 2020). Such developments have not yet been possible in the Arctic, as large-sample databases of smaller watersheds are not yet available.

Here we present an integrated pan-ARctic CAtchments summary DatabasE (ARCADE) of >40,000 catchments, including small and medium-sized watersheds, draining into the Arctic Ocean. These watersheds, delineated at a high-resolution (90 m), are coupled with comprehensive information from various geospatial environmental, climatic, and physiographic datasets with pan-Arctic coverage. This publication aims to provide a high-resolution geographical register, relevant to those studying environmental and climatic changes in relation to Arctic catchment hydrology and biogeochemistry.

## 2. Methods

### 2.1 Spatial extent and projection

The ARCADE database encompasses all major and minor drainage basins that are considered part of the pan-Arctic watershed, draining into the Arctic Ocean and surrounding seas. More specifically, this includes all watersheds with a Strahler order of five (i.e. at least five hierarchical branching orders) or larger that drain into the Arctic Ocean as well as basins draining into the Bering Sea, and North of the Yukon River outlet with inclusion of the Yukon River. This follows the pan-Arctic watershed definition as defined by McGuire et al. (2009), with an area of $20.4 \times 10^6$ km$^2$, including the Canadian Archipelago, Greenland, and Hudson Bay (fig. 1). The data presented here have been transformed/re-projected to WGS84 / NSIDC EASE-Grid 2.0 North (EPSG:6931) projection, an equal-area projection system designed for gridding and small-scale digital mapping for environmental sciences in the Northern Hemisphere (Brodzik et al., 2014).

### 2.2 Watershed delineation

#### 2.2.1 Digital elevation model (DEM)

Terrain parameters such as altitude, slope, aspect, topographic position index, and Slope Length and Steepness factor (LS-factor) (Renard et al., 2017) were derived and calculated from Copernicus DEM GLO-90, a high-quality global 90-m resolution digital elevation model provided by the European Space Agency (ESA, 2021). The Copernicus DEM was accessed on 10 September 2021. For computational practicality, we chose the 90 m resolution product rather than the 30-m resolution product. The latter could be used for future version updates of the ARCADE database. However, we deem the 90-m



resolution sufficiently detailed for our purposes (gaining insights in drainage areas on a pan-Arctic scale). We constrain the number of catchments in the database by using Strahler order 5 as the minimum outlet order and 1 km$^2$ catchment area as a lower threshold value (see next paragraph). A higher resolution DEM would not necessarily make for a better delineation. Moreover, most of the datasets used to link to the catchment areas have lower resolutions than 90 m. We are aware that, at any given resolution, the relative

error regarding catchment delineation increases when looking at smaller watersheds. Yet, at our chosen resolution, we conclude there to be a reasonable tradeoff between efficiency and error.

**2.2.2 Hydrological DEM conditioning and watershed extraction**

The DEM was hydrologically conditioned (a.k.a. pit filling) before deriving flow direction, flow accumulation, Strahler order, watershed delineations, and topographic wetness index. This was done

using the *r.hydrodem* module (Lindsay & Creed, 2005) in GRASS GIS (Neteler et al., 2012).

We delineated the watersheds at 90-meter resolution for subdivisions of the pan-Arctic landmass using the hydrologically-conditioned DEM. This subdivision was necessary because processing the DEM in one piece was computationally too intensive. Delineation was done using SAGA GIS (Conrad et al.,

2015) using the module *Channel Network and Drainage Basins*. A lower threshold of Strahler order 5 was chosen to constrain watershed generation, i.e., only watersheds of streams with Strahler order 5 or higher at the outlet were delineated. This threshold was necessary to limit the number of watersheds in the final product and to ensure that only watersheds with actual streams were included. Another consideration was that as the watershed area approaches the DEM's source resolution, the relative

accuracy decreases. Subdivisions of the pan-Arctic watersheds were combined into one dataset of all watersheds that drain into the AO (i.e., upstream areas of outlets at the AO). A known limitation of DEM-derived watershed delineation is that the algorithm struggles to find the actual channels and ridges in flat terrain. Since we are mostly interested in the drainage area, rather than channel location, errors in channels were tolerated more than errors in catchment boundaries. Small flat catchments (area < ~10 km$^2$ and slope

< ~0.1° and mainly in fluvial deltas) are most prone to error, which is why we advise users to be critical when using these delineations for local purposes.

**2.3 Environmental data**

All variables are described in supplementary table S1. Elaborated explanations are provided below.

**2.3.1 Climatological data**

Climatological data were extracted from the ERA5-Land Monthly Averaged – ECMWF Climate Reanalysis dataset (Muñoz-Sabater et al., 2019) using Google Earth Engine ("Planetary-scale geospatial analysis for everyone") (Gorelick et al., 2017). This dataset has a spatial resolution of 11132 meters and consists of 50 bands containing climatological variables related to temperature, precipitation, evaporation, heat fluxes, wind, and vegetation. Minimum, maximum, mean, standard deviation, and median annual

values of a subset of these variables (a complete overview of all variables is available in S1) were calculated for each watershed from 01-01-1990 to 31-12-2019. In the case of pixels falling partly within



the geometry of a watershed, the value is weighted by the fraction of each pixel that falls within the geometry. Precipitation, evaporation, and runoff totals were accumulated and averaged over the 30-year period (i.e., the mean annual total of each of these variables was calculated). For snow, we calculated the
30-year average maximum monthly snow depth (m), snow cover (%), snowmelt (m d$^{-1}$), and snowfall (m d$^{-1}$) (i.e., based on the month with the highest value of each year). As an example: A given watershed can have an average snowmelt of 0.021 m d$^{-1}$. This means that on average the month with the most snowmelt has an average snowmelt rate of 0.021 m d$^{-1}$, which can be seen as an indicator of intensity of the melting season.

We also tested for trends using Sen's slope estimator for the same period. Sen (1968) calculates the slope as:

$$Q_i = \frac{(x_j - x_i)}{j - i}, i = 1, 2, 3, \dots, N \qquad (1)$$


Where $x_j$ and $x_i$ are records at time $j$ and $i$ ($j > i$). With $n$ data records in a timeseries, the number of slope estimates equals $N = n(n-1)/2$. $Q_i$ then follows from calculating the median of all the slope estimators. We chose to calculate these statistics on monthly data for temperature variables, while for snow-related variables, we only selected the winter months (November – April) and for evaporation-related variables
only the summer months (June – September).

### 2.3.2 Physiographic data

### Catchment properties

Basic catchment properties include minimum, maximum, mean, standard deviation and median of elevation (meters), slope (degrees), and aspect (degrees). Furthermore, we included centroid latitude
(degrees), Gravelius index (watershed perimeter divided by the perimeter of a circle that has the same area; unitless), watershed perimeter (kilometers) and watershed area (square kilometers).

### Soil properties

SoilGrids is a globally consistent dataset that predicts soil properties and classes at 250-meter resolution (Poggio et al., 2021). The ARCADE database includes aggregated fractional coverage of the most likely
soil class according to the World Reference Base (WRB) classification system (IUSS Working Group WRB, 2014). Additionally, we calculated watershed minimum, maximum, mean and standard deviation for soil organic carbon (SOC) content (dg kg$^{-1}$), organic carbon density (dg dm$^{-3}$), nitrogen content (cg kg$^{-1}$), coarse fragments volumetric content (per 10000), sand, silt and clay content (g kg$^{-1}$), soil bulk density (cg cm$^{-3}$), all at depth intervals of 0-5 cm, 5-15 cm, 15-30 cm, 40-60 cm, 60-100 cm. Additionally,
the modeled organic carbon stocks (OCS) (t ha$^{-1}$) in the upper 30 cm of the soil were added. The latter was also accumulated by us into watershed OCS (gt).



### Landcover class fractional cover

Watershed land cover fractional coverage was obtained from ESA WorldCover 10m v100 (Zanaga et al., 2021). This classifies the land surface at 10-meter resolution into 11 classes: trees, shrubland, grassland,
cropland, built-up areas, barren/sparse vegetation, snow and ice, open water, herbaceous wetland, mangroves, moss and lichen.

### Landform class fractional coverage

Another useful characterization parameter for watersheds is the fractional coverage of landforms. We chose to use a landform classification scheme proposed by Theobald et al. (2015). Their classification
scheme maps ecologically-relevant landforms (see supplementary tables included in dataset S1), which we deem of particular interest in characterizing a catchment, for instance to indicate sensitivity to occurrence of abrupt permafrost thaw.

### Burned area fraction coverage

Burned area fraction for each watershed over the period 2012 - 2022 was calculated from MODIS
FireCCI5, a monthly global 250-meter spatial resolution burn scar classification product (Chuvieco et al., 2018). We selected and summarized relatively recent (<10 yr) annual fire scars as they are most likely to have an ongoing and lasting effect on watershed biogeochemistry.

### Permafrost extent

Permafrost fraction pixel cover was converted from the permafrost extent by Obu et al. (2019) and
converted into watershed area fractional coverage per permafrost coverage type. The used product has a spatial resolution of 1 kilometer and a temporal range from 2000 – 2016. Continuous permafrost is classified as a pixel area coverage of 90-100%, discontinuous permafrost as 50-90%, sporadic permafrost as 10-50%, and isolated patches of permafrost as 0-10%.

### Active layer thickness

Recently published high-resolution estimates of active layer thickness (ALT) (Ran et al., 2021) were summarized for each watershed. The source dataset has a 1-kilometer resolution for the period of 2000-2016. The authors generated the data by combining large amounts of field data combined with multisource geospatial remote sensing data into a statistical learning model. It has bias = 2.71±16.46 cm and RMSE = 86.93±19.61 cm for ALT.

### Glacial fractional coverage

Glacial coverage was calculated by combining two datasets: Global Land Ice Measurements from Space (GLIMS) from which we used the latest available snapshot as of September 14, 2021, for the glacial extent (Kargel et al., 2014) and the Greenland Ice & Ocean Mask from the Greenland Mapping Project





(GIMP) which contains a 15-meter resolution land ice mask for the Greenland ice sheet (Howat et al.,
2014). We resampled the combined datasets to a 250-meter resolution grid to calculate fraction glacial
coverage for each watershed.

**Surface water fractional coverage**

A high-resolution water mask, JRC Global Surface Water Mapping Layers, v1.3, (30 meter) (Pekel et al.,
2016) was used to calculate fractional watershed area coverage. The conditions for presence of water
were determined by occurrence of 50% of the time between 1984 and 2020.

**Vegetation index**

The summarized statistics of the normalized difference vegetation index (NDVI) and the Sen slope of
NDVI were calculated using MOD13A1.006 Terra Vegetation Indices 16-Day Global 500m (Didan,
2015; accessed: 1 April 2022). This dataset is MODIS derived and has a 500-meter resolution. We
selected all available data in the month of August from 2000 to 2021.

**Topographic wetness index**

As an indicator of terrain wetness, we used SAGA wetness index (Böhner & Selige, 2006), a modified
topographic wetness index that is based on Moore et. al (1993). The indicator uses topography to
differentiate catchments dominated by wetland terrain versus more well-drained terrain.

**LS-factor**

Slope Length and Steepness factor (LS-factor) is a factor used in the Universal Soil Loss Equation (USLE)
(Renard et al., 2017) that serves as a predictor of soil loss ratio as a function of slope length and steepness.
The LS-factor was calculated using SAGA GIS tool Module LS-factor which uses specific catchment
area (SCA) as a substitute of slope length (Böhner & Selige, 2006).


**Tasseled-cap trend index of visible spectra**

As an indicator for changes in wetness (TCW), greenness (TCG) and brightness (TCB) (indicative of bare
soil) we the slope of tasseled cap indices derived from Landsat visible spectra images as provided by
Nitze et al. (2018). The minima, maxima, and average of these pixel-based slopes were calculated for
each watershed.



## 3. Results and discussion

### 3.1 Database inventory

The database consists of 47054 watersheds ranging in size from 1 km$^2$ to 3.1 x 10$^6$ km$^2$ (Ob' watershed).
We will refer to four groups of watersheds based on size (fig. 1, tables 1-5) because our work focuses on
inventorying watersheds of all sizes and highlights the contrasts between the larger well-studied rivers
and smaller rivers. The first group consists of 'The Big Six' (Ob', Yenisey, Lena, Mackenzie, Yukon,
Lena Rivers) and one major watershed draining into the Hudson Bay (Nelson River). Therefore, the 'The
Big Six' becomes 'The Big Seven', abbreviated as BS. Then, 'The Middle Nine' (MN) consists of the
Severnaya Dvina, Indigirka, Pechora, Olenek, Thelon, Yana, Khatanga, Pyasina and Taz Rivers. We then
split the remaining watersheds into areas greater than 1000 km$^2$ (yet smaller than the MN), which we
named "The Pan-Arctic 1000's" (PAT), and watersheds smaller than 1000 km$^2$, which we named the
"The Pan-Arctic Small watersheds" (PAS).

The BS account for 50% of the total AO watershed area, while watersheds under 1000 km$^2$ (PAS) account
for only 9% of the entire area. However, these small watersheds are much more abundant, and their
landmass is more directly connected to the Arctic Ocean than the BS. Since large parts of the BS
watersheds reach into low latitudes (~60% of their watershed areas are located south of 60° North, or
93% south of the Arctic Circle), the mean annual air temperature in these watersheds is higher compared
to the rest of the pan-Arctic watershed (table 3), influencing mean permafrost coverage, active layer
thickness (ALT), and occurrence of ice-wedge polygon terrain (table 4, 5, 6). These permafrost-related
watershed properties are susceptible to change under climate warming trends and play a central role in
Arctic watershed hydrology. Our database shows that the MN, PAT, and PAS watersheds have been
warming much faster than the BS (table 7), highlighting the need for more research on these smaller
northern watersheds.

### 3.2 Data coverage

The hydrological functioning of many small catchments in the Arctic remains uncertain. Therefore, our
database provides a set of catchment properties to help address these uncertainties. Basic topographical
catchment metrics such as area, elevation, catchment slope, mean aspect, LS-factor, and TWI are available
for all recorded catchments. Due to their resolution and extent, some of the other aggregated datasets have
lower coverage. Most notably, ERA5-Land data has ~87% spatial coverage in the database. Most omitted
watersheds are small coastal watersheds that were less than 50% covered by a cell of the ERA5-Land
dataset. The same holds for the ALT and SoilGrids data which cover about ~82% and ~92% of all
watersheds, respectively. For all aggregated data sources, >80% of watersheds are covered.

### 3.3 Data quality assessment and limitations

The ARCADE database is the first published 90-meter resolution dataset of watersheds draining into the
Arctic Ocean. A few unavoidable errors occurred during the watershed delineation. Errors most
commonly arise in flat terrain where flow-routing algorithms struggle to determine the correct flow




direction, which troubles the watershed border definition. To deal with this flow direction problem, we used an internal SAGA function to artificially maintain a minimal channel slope by slightly altering the 280 DEM. This minimal slope function effect is visually detectable in small deltas and floodplains where watershed borders sometimes appear less accurate than in steep, well-defined terrain. Additionally, this flow path uncertainty in flat terrain caused some errors in approximating the locations of coastal outlets. Given the high DEM resolution, these errors are generally in the order of meters rather than kilometers. This could be improved in future versions by 'burning' outlets and channels, for example derived from 285 satellite imagery, into the DEM.

Our cut-off value in defining a river catchment (outlet Strahler order 5; minimum area of 1 km$^2$) leads to the omission of areas that lie within the pan-Arctic drainage basin but are outside our database's scope (i.e. so called wolf-tooth patches, remaining coastal areas in between catchments). However, we estimate 290 the summarized area to be less than 1% of the total pan-Arctic watershed area. The strength of this database lies in the large spatial extent, its novelty and the range of spatially-explicit variables coupled to the delineated catchments. We therefore advise using this database to target specific (groups of) catchments and making comparisons among those to gain insight into spatial patterns, and localization of target areas for further research.

**3.4 Pan-Arctic watersheds properties**

ARCADE provides 103 variables with catchment properties divided over 353 columns (including statistics), showcasing a wide variability as well as spatial resemblances of catchments in the pan-Arctic drainage basin. Additionally, we provide summaries of the most important properties for the BS, MN, PAT, and PAS, both as a whole and on a regional basis (i.e., North America, Greenland, and Eurasia).

**Physiographic features**

Basic catchment-scale topographical information can be used to categorize watershed types and estimate their runoff, sediment transport regimes, and biogeochemical constituents. As an example, Connoly et al. (2018) found strong negative correlations between catchment slope and DOM and $NO_3$- concentrations in Arctic watersheds. According to the data presented here, PAS watersheds have on average the highest 305 mean catchment slope. This is partially because Greenlandic small coastal watersheds are mountainous (table 3 and table 4). Eurasia and North America's proportion of PAT, on the other hand, consist of relatively low elevation, flat terrain (mean slope Eurasia: ~3.1 ±1.54°, North America: ~2.4 ±1.53°).. The PAS watersheds are underlain mainly by continuous permafrost and feature wetland-type landcover (Eurasia: ~27% wetland North America: ~14% wetland) as opposed to BS (Eurasia 4% wetland, North 310 America 1% wetland), with a high area-fraction of surface water (Eurasia: ~6% water, North America: ~8% water). Because of their permafrost coverage (mostly continuous), PAS watersheds are more likely to feature IWP terrain (37% IWP terrain in PAS as opposed to 1% IWP terrain in BS). Another noteworthy property of PAS watersheds is that, on average, they feature higher OC stocks (Eurasia: ~88 t ha$^{-1}$, Greenland: ~87 t ha$^{-1}$, North America: ~71 t ha$^{-1}$) than more commonly studied catchments (BS: ~64.5 t 315 ha$^{-1}$, MN: ~69.5 t ha$^{-1}$). Additionally, Greenland stands out in most aspects, with a relatively high mean





catchment slope (~7.5°), elevation (532 m amsl), and glacial coverage (77%) (table 4 and table 5). This distinction in basic characteristics most likely distinguishes the lateral flux characteristics of Greenlandic watersheds from the rest of PAS. Greenland also includes several (149 out of 929) PAT catchments which are largely (>80% of their area) covered by the Greenlandic Ice Sheet. Since principles of watershed
hydrology do not apply to ice sheets or glaciers, we advise users of this database to take note of the presence of these 'ice sheet watersheds' in the database. A solution to circumvent these watersheds is filtering by fraction ice coverage to a value aligned with the study goals.

**Climatological properties**

Since MN, PAT and PAS are on average located in higher latitudes, these watersheds are colder than the
BS (BS: -2.9 °C, MN: -10.5 °C, PAT: -9.1 °C, PAS: -10.4 °C) (table 3). While for the BS, the Eurasian watersheds are the coldest, the opposite is true for PAS (PAS of Eurasia: -4.7° C, North America: -11.0° C. This is partially because the Gulf Stream warms smaller coastal watersheds of western Eurasia, but there might also already be some effect of temperature increase which has been greatest in the Eurasian PAS (+3.4° C; table 7). Annual precipitation, mean annual runoff, and the mean increase in precipitation
over the past 30 years are highest in PAS and PAT (i.e., smaller watersheds) (table 7).

**Pan-Arctic trends in data**

We provide this database as a basis to explore the vast number of watersheds outside the BS and MN that have previously been lumped into a single 'unknown'. As a result, they have been underappreciated in
terms of their contribution to the pan-Arctic lateral flux budget and their potential sensitivity to climate change as opposed to their bigger siblings. While continuing the scientific focus on large catchment studies (BS) in the Arctic remains vital, we suggest to, in parallel, strongly increase focus pan-Arcticat small catchments situated entirely at high latitudes. These catchments are experiencing the greatest climatic warming while also storing large quantities of soil carbon in landscapes that are especially prone
to degradation of permafrost (i.e., IWP terrain) and associated hydrological regime shifts. Using our database these and many other variables are now quantified and made spatially explicit (fig. 3, 4).

**4. Database availability**

Data are publicly available on https://dataverse.nl/dataset.xhtml?persistentId=doi:10.34894/U9HSPV under Creative Commons License Attribution 4.0 International (CC BY 4.0).

**5. Outlook and future development**

ARCADE is the first aggregated database of pan-Arctic river catchments that includes small watersheds at a high resolution. The publication of this database is a necessary step toward more integrated monitoring of the pan-Arctic watershed. An important addition in the following version will be discharge data and derived seasonality (and changes therein) from the RADR database (Feng et al., 2021), which
recently greatly advanced understanding of discharge in smaller arctic rivers. Another important future



addition will be the delineations of subbasins and data on river biogeochemistry that is available, albeit non-uniformly and largely unaggregated throughout literature. When numerous valuable datasets from various scientific disciplines are merged, it will be possible to better understand the Arctic's changing hydrology and biogeochemistry. This should allow the scientific community to form new hypotheses that

direct scientific efforts to specific regions and processes that may have remained under the radar.



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





**Table 1 Summary statistics of the pan-Arctic watersheds database focused on permafrost. Note that this summary excludes watersheds that are fully covered by glaciers/icesheets.**

| Group* | n | Total area (km²) | % of total area | Mean elevation (m amsl*) | Mean slope (°) | Permafrost | | | Ice-wedge terrain | Mean ALT* (cm) | MAAT* (°C) |
|---|---|---|---|---|---|---|---|---|---|---|---|
| | | | | | | Continuous | Discontinuous | Sporadic | | | |
| BS | 7 | $1.31{\times}10^7$ | 50% | 289 | 2.5 | 19% ±26.8% | 19% ±13.6% | 14% ±11.2% | 1% | 126 ±29 | -2.9 ±3.7 |
| MN | 9 | $2.31{\times}10^6$ | 12% | 306 | 2.0 | 57% ±46.8% | 11% ±17.1% | 8% ±14.1% | 10% | 94 ±14 | -10.5 ±7.6 |
| PAT | 929 | $6.09{\times}10^6$ | 29% | 212 | 1.6 | 48% ±46.1% | 12% ±28.5% | 9% ±23.5% | 31% | 98 ±42 | -10.4 ±6.1 |
| PAS | 45124 | $2.23{\times}10^6$ | 9% | 112 | 3.4 | 57% ±46.4% | 9% ±26.5% | 7% ±21.9% | 37% | 93 ±45 | -9.1 ±5.8 |
| **Total** | 46069 | $2.37{\times}10^7$ | 100% | 230 | 2.4 | 45% ±84.8% | 13% ±44.6% | 9% ±36.8% | 20% | 103 ±32 | -8.2 ±5.8 |

*Abbreviations stand for the grouped watersheds by area, Big Seven (BS), Middle Nine (MN), Pan-Arctic Thousands (PAT), Pan-Arctic Small watersheds (PAS); amsl stands for 'above mean sea level'; ALT stands for 'Active Layer Thickness'; MAAT stands for 'Mean Annual Air Temperature'.





**Table 2 Watershed topographic properties summarized by group (classification based on area) and relevant (sub)continent. Note that _n_ stands for the number of watersheds that have these climatological parameters attached to them.**

| Group* | Continent | n | Max. mean slope (°) | Mean slope (°) | Mean area (km²) | Total area (km²) | Mean elevation (m) | Water (%) | Ice (%) | Mean TWI* | Mean LS* |
|---|---|---|---|---|---|---|---|---|---|---|---|
| BS | Eurasia | 4 | 4.9 | 3.1 ±1.54 | 2283485 | 9133939 | 290 ±103.1 | 2% | 0% | 6.6 ±1.29 | 3.1 ±1.57 |
| BS | North America | 3 | 3.7 | 2.5 ±1.53 | 1313306 | 3939919 | 290 ±64.3 | 7% | 0% | 6.8 ±2.25 | 2.7 ±1.93 |
| MN | Eurasia | 8 | 4.7 | 1.8 ±1.59 | 259073 | 2072580 | 154 ±122.2 | 3% | 0% | 6.9 ±1.90 | 1.7 ±1.86 |
| MN | North America | 1 | 0.6 | 0.6 ±0.00 | 238539 | 238539 | 135 ±0.0 | 22% | 0% | 8.7 ±0.00 | 0.2 ±0.00 |
| PAT | Eurasia | 285 | 8.8 | 1.6 ±1.71 | 6444 | 1836498 | 97 ±115.8 | 7% | 1% | 6.9 ±1.95 | 1.5 ±2.39 |
| PAT | Greenland | 140 | 10.7 | 2.3 ±2.42 | 4781 | 669384 | 532 ±259.2 | 1% | 77% | 4.8 ±1.45 | 3.2 ±3.97 |
| PAT | North America | 504 | 9.9 | 1.5 ±1.72 | 7105 | 3581003 | 133 ±127.0 | 10% | 5% | 6.2 ±1.76 | 1.4 ±2.26 |
| PAS | Eurasia | 14269 | 27.0 | 3.1 ±4.10 | 45 | 645781 | 80 ±108.2 | 6% | 6% | 5.6 ±1.88 | 3.9 ±6.64 |
| PAS | Greenland | 7848 | 28.9 | 7.5 ±5.04 | 40 | 310800 | 258 ±179.8 | 2% | 29% | 3.7 ±0.98 | 11.7 ±9.88 |
| PAS | North America | 22996 | 23.1 | 2.4 ±2.99 | 55 | 1272918 | 81 ±112.3 | 8% | 4% | 5.2 ±1.58 | 2.7 ±4.90 |
| **Total** | Pan-Arctic | 46058 | 12.2 | 2.6 ±8.37 | 411287 | 23701361 | 205 ±427.8 | 7% | 12% | 6.1 ±5.13 | 3.2 ±14.21 |

\* Abbreviations stand for the grouped watersheds by area, Big Seven (BS), Middle Nine (MN), Pan-Arctic Thousands (PAT), Pan-Arctic Small watersheds (PAS); TWI stands for 'Topographic Wetness Index' and is based on the SAGA Wetness Index Tool; LS stands for 'Slope Steepness and Length Factor'.

**Table 3 Watershed permafrost properties summarized by group (based on area) and relevant (sub)continent. Note that _n_ stands for the number of watersheds that have these climatological parameters attached to them.**

| Group* | Continent | n | Continuous permafrost | Discontinuous permafrost | Sporadic permafrost | IWP* terrain | OCS$_{0-30cm}$ (t ha$^{-1}$) | ALT* mean (cm) |
|---|---|---|---|---|---|---|---|---|
| BS | Eurasia | 4 | 30% ±31.8% | 18% ±12.5% | 10% ±6.7% | 2 ±4.3% | 67 ±3.7 | 128 ±29.3 |
| BS | North America | 3 | 4% ±4.6% | 20% ±17.9% | 20% ±15.0% | 3 ±2.2% | 62 ±4.7 | 118 ±29.9 |
| MN | Eurasia | 8 | 59% ±49.6% | 7% ±11.6% | 8% ±15.1% | 10 ±12.1% | 78 ±14.5 | 87 ±12.4 |
| MN | North America | 1 | 40% ±0.0% | 46% ±0.0% | 7% ±0.0% | 39 ±0.0% | 61 ±0.0 | 112 ±0.0 |
| PAT | Eurasia | 285 | 43% ±47.5% | 18% ±35.2% | 10% ±25.4% | 40 ±39.9% | 83 ±12.9 | 96 ±56.9 |
| PAT | Greenland | 140 | 15% ±24.7% | 2% ±6.8% | 0% ±1.7% | 4 ±8.5% | 92 ±13.9 | 105 ±49.3 |
| PAT | North America | 504 | 59% ±45.3% | 12% ±27.1% | 11% ±25.0% | 37 ±38.1% | 71 ±11.2 | 96 ±28.0 |
| PAS | Eurasia | 14279 | 41% ±46.7% | 11% ±29.4% | 6% ±20.9% | 38 ±42.9% | 88 ±13.1 | 89 ±55.4 |
| PAS | Greenland | 7848 | 35% ±41.4% | 11% ±26.2% | 7% ±21.2% | 13 ±28.2% | 87 ±14.4 | 113 ±63.4 |
| PAS | North America | 22996 | 74% ±40.8% | 8% ±24.6% | 7% ±22.7% | 45 ±43.9% | 71 ±12.3 | 87 ±24.7 |
| **Total** | Pan-Arctic | 46068 | 40% ±118.2% | 15% ±69.2% | 9% ±56.3% | 23% ±88.6% | 76 ±35.5 | 103 ±126.7 |

\*Abbreviations stand for the grouped watersheds by area, Big Seven (BS), Middle Nine (MN), Pan-Arctic Thousands (PAT), Pan-Arctic Small watersheds (PAS); IWP stands for 'Ice Wedge Polygon'; ALT stands for 'Active Layer Thickness'.





**Table 4 Watershed land-cover type and properties summarized by group (classification based on area) and relevant (sub)continent. Note that *n* stands for the number of watersheds that have these climatological parameters attached to them.**

| Group* | Continent | n | Trees | Shrub | Grassland | Cropland | Built-up | Barren | Snow/ice | Water | Wetland |
|--------|-----------|---|-------|-------|-----------|----------|----------|--------|----------|-------|---------|
| BS | Eurasia | 4 | 56 ±19.2% | 1 ±0.9% | 24 ±10.1% | 3.6 ±6.53% | 0.1 ±0.11% | 2 ±1.2% | 0 ±0.0% | 3 ±0.7% | 4 ±3.9% |
| BS | North America | 3 | 47 ±9.1% | 5 ±0.6% | 23 ±7.6% | 9.0 ±14.61% | 0.1 ±0.16% | 2 ±1.1% | 1 ±1.0% | 8 ±4.9% | 1 ±0.5% |
| MN | Eurasia | 8 | 53 ±26.6% | 0 ±0.2% | 23 ±17.3% | 0.1 ±0.33% | 0.0 ±0.03% | 1 ±1.5% | 0 ±0.0% | 4 ±2.8% | 8 ±8.2% |
| MN | North America | 1 | 6 ±0.0% | 0 ±0.0% | 51 ±0.0% | 0.0 ±0.00% | 0.0 ±0.00% | 1 ±0.0% | 0 ±0.0% | 26 ±0.0% | 1 ±0.0% |
| PAT | Eurasia | 285 | 12 ±22.2% | 0 ±0.4% | 36 ±20.8% | 0.1 ±0.32% | 0.0 ±0.08% | 4 ±8.7% | 2 ±7.7% | 8 ±7.4% | 21 ±22.8% |
| PAT | Greenland | 140 | 0 ±0.1% | 0 ±0.0% | 4 ±7.6% | 0.0 ±0.00% | 0.0 ±0.00% | 8 ±10.1% | 77 ±27.8% | 2 ±2.8% | 0 ±0.1% |
| PAT | North America | 504 | 8 ±17.3% | 3 ±8.4% | 25 ±24.0% | 0.0 ±0.02% | 0.0 ±0.01% | 12 ±14.4% | 5 ±17.3% | 12 ±9.5% | 5 ±10.4% |
| PAS | Eurasia | 14279 | 10 ±21.8% | 0 ±0.5% | 29 ±30.4% | 0.2 ±1.88% | 0.1 ±1.34% | 10 ±21.3% | 6 ±18.1% | 7 ±11.1% | 27 ±33.2% |
| PAS | Greenland | 7848 | 0 ±0.5% | 0 ±0.0% | 14 ±22.3% | 0.0 ±0.00% | 0.0 ±0.36% | 23 ±22.0% | 30 ±33.6% | 5 ±7.8% | 1 ±3.8% |
| PAS | North America | 22997 | 2 ±10.9% | 1 ±5.6% | 16 ±23.2% | 0.0 ±0.05% | 0.0 ±0.13% | 30 ±28.3% | 5 ±16.7% | 10 ±11.7% | 14 ±22.8% |
| **Total** | Pan-Arctic | 46069 | 19% ±50.4% | 1% ±10.1% | 25% ±59.0% | 1% ±16.1% | 0% ±1.4% | 9% ±46.1% | 13% ±53.6% | 8% ±22.5% | 8% ±48.5% |

*Abbreviations stand for the grouped watersheds by area, Big Seven (BS), Middle Nine (MN), Pan-Arctic Thousands (PAT), Pan-Arctic Small watersheds (PAS).

**Table 5 Watershed climatological properties summarized by group (classification based on area) and relevant (sub)continent. Note that *n* stands for the number of watersheds that have these climatological parameters attached to them.**

| Group* | Continent | n | T min. (°C) | T max. (°C) | T mean (°C) | ΔT mean (°C 30y⁻¹) | ET mean (mm yr⁻¹) | P mean (mm yr⁻¹) | ΔP mean (mm yr⁻¹) | Q mean (mm yr⁻¹) | max. Snow depth (m) | max. Snowmelt (mm d⁻¹) |
|--------|-----------|---|-------------|-------------|-------------|--------------------|--------------------|-------------------|--------------------|-------------------|----------------------|------------------------|
| BS | Eurasia | 4 | -20.0 ±4.38 | 9.0 ±4.69 | -5.5 ±4.95 | 1.7 ±0.65 | 325 ±74.4 | 513 ±66.3 | 1.2 ±1.60 | 193 ±30.9 | 0.7 ±0.08 | 4.0 ±0.37 |
| BS | North America | 3 | -16.3 ±3.02 | 12.2 ±4.13 | -2.1 ±3.38 | 1.5 ±0.96 | 377 ±112.0 | 541 ±50.7 | 1.1 ±1.79 | 178 ±53.6 | 0.7 ±0.35 | 3.3 ±0.97 |
| MN | Eurasia | 8 | -22.3 ±5.19 | 5.7 ±4.93 | -8.3 ±5.62 | 2.8 ±0.88 | 243 ±73.9 | 515 ±150.4 | 1.0 ±1.30 | 274 ±108.2 | 0.7 ±0.19 | 4.9 ±1.84 |
| MN | North America | 1 | -21.8 ±0.00 | 5.8 ±0.00 | -9.0 ±0.00 | 1.8 ±0.00 | 255 ±0.0 | 421 ±0.0 | 0.3 ±0.00 | 170 ±0.0 | 0.6 ±0.00 | 3.4 ±0.00 |
| PAT | Eurasia | 285 | -18.3 ±6.28 | 5.8 ±4.77 | -6.4 ±5.97 | 3.4 ±1.42 | 192 ±69.9 | 601 ±376.6 | 3.7 ±3.50 | 403 ±350.4 | 1.6 ±3.94 | 6.1 ±3.04 |
| PAT | Greenland | 140 | -26.0 ±4.08 | -4.8 ±4.33 | -15.5 ±4.25 | 2.0 ±0.38 | 24 ±37.8 | 561 ±454.1 | 1.8 ±7.77 | 67 ±123.5 | 29.6 ±7.81 | 1.0 ±1.63 |
| PAT | North America | 504 | -21.8 ±4.44 | 3.1 ±6.37 | -10.0 ±5.18 | 2.5 ±0.75 | 200 ±99.3 | 462 ±229.0 | 1.8 ±3.46 | 257 ±183.2 | 2.9 ±6.89 | 4.6 ±2.14 |
| PAS | Eurasia | 14272 | -13.8 ±7.89 | 4.7 ±5.77 | -4.7 ±6.88 | 3.4 ±1.79 | 199 ±149.7 | 794 ±591.9 | 3.6 ±5.29 | 564 ±530.3 | 5.7 ±10.85 | 6.3 ±3.57 |
| PAS | Greenland | 7844 | -18.3 ±5.33 | 0.8 ±5.35 | -9.0 ±5.26 | 2.3 ±0.56 | 65 ±76.1 | 790 ±578.0 | 2.9 ±12.70 | 334 ±359.8 | 20.1 ±14.70 | 4.5 ±4.40 |
| PAS | North America | 22989 | -20.6 ±4.76 | -0.2 ±5.22 | -11.0 ±4.54 | 2.5 ±0.83 | 160 ±86.4 | 401 ±204.1 | 1.6 ±3.37 | 225 ±169.5 | 4.0 ±8.67 | 4.5 ±2.09 |
| **Total** | Pan-Arctic | 46050 | -19.9 ±15.63 | 4.2 ±15.31 | -8.2 ±15.61 | 2.4 ±3.01 | 204 ±274.6 | 560 ±1075.2 | 1.9 ±17.11 | 266 ±791.5 | 6.7 ±23.09 | 4.3 ±7.58 |

*Abbreviations stand for the grouped watersheds by area, Big Seven (BS), Middle Nine (MN), Pan-Arctic Thousands (PAT), Pan-Arctic Small watersheds (PAS); T stands for temperature, ET stands for total evapotranspiration, P stands for precipitation and Q stands for runoff. ΔT mean and ΔP mean are calculated from the Sen slope of the monthly mean Temperature and Precipitation over the period 1999 – 2019 (mo⁻¹) multiplied by the number of months.





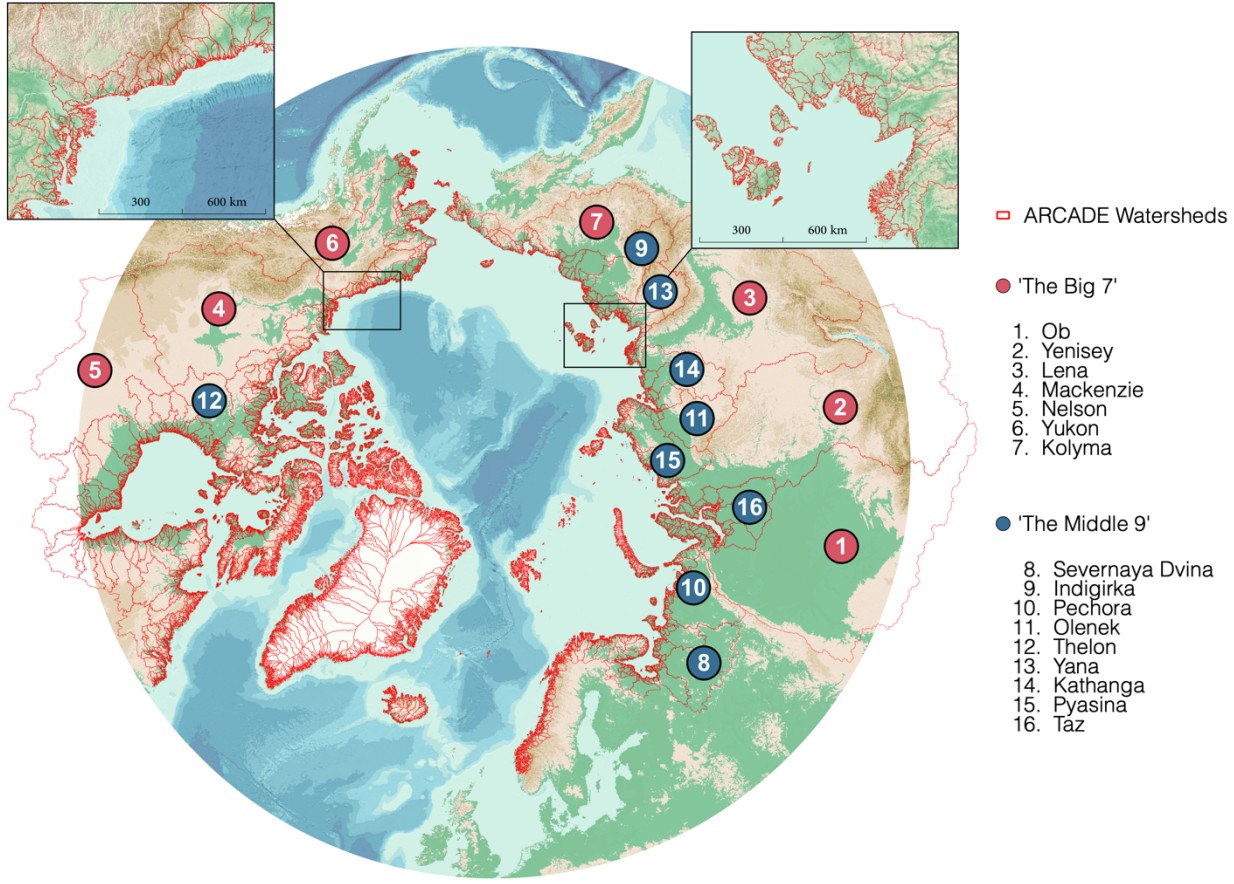

**Figure 1. Circumpolar map of all ARCADE watersheds, 1 km² and larger, Strahler order 5 and higher, at 90-meter resolution with insets of the Southern Beaufort Sea region (upper left) and the Laptev Sea coast, including the New Siberian Islands (upper right). (Background map: International Bathymetric Chart of the Arctic Ocean V4.0 (IBCAO) (Jakobsson et al., 2020)).**

Earth System
Science
Data

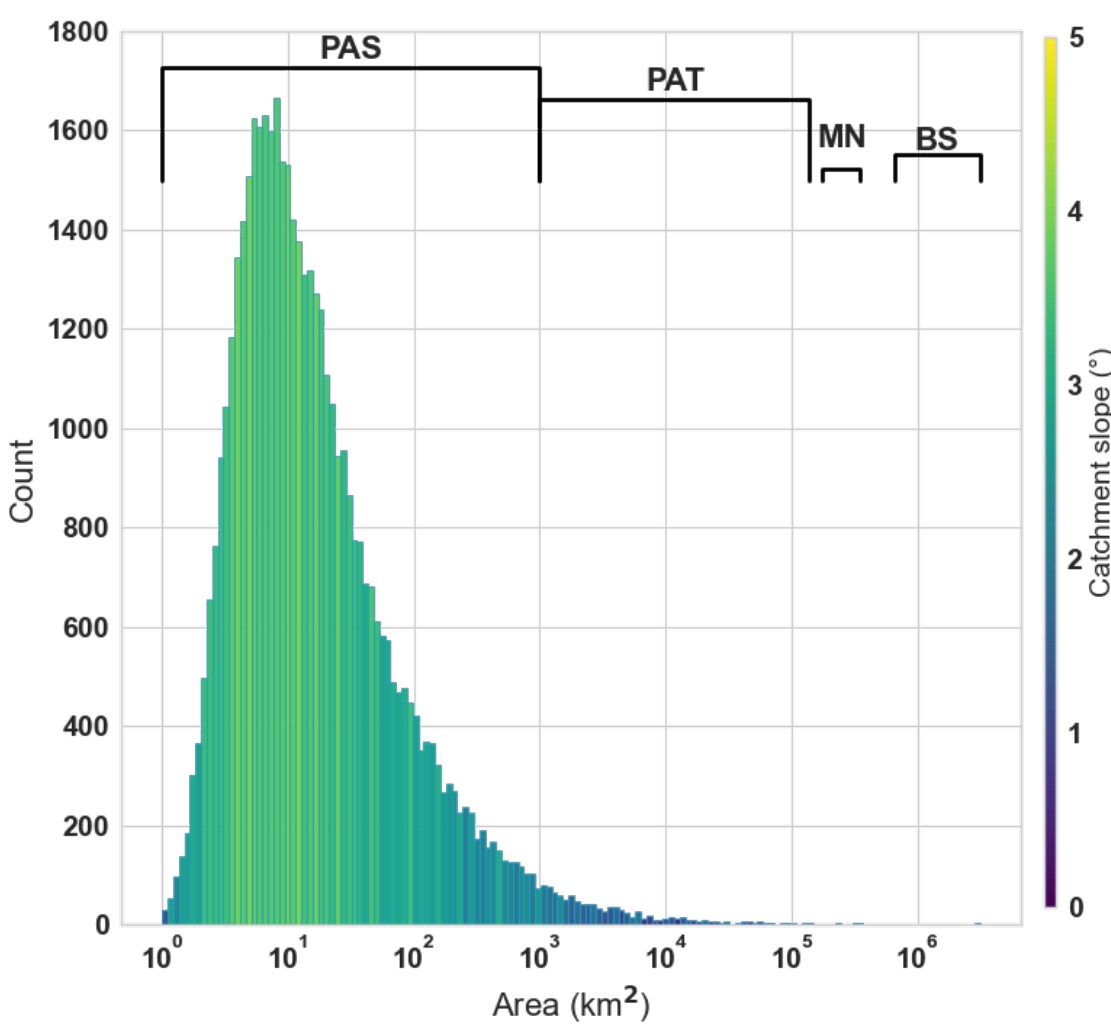

**Figure 2. The distribution of watershed areas in the pan-Arctic watersheds database and the range of the four groups that classify based on watershed area. 'BS' stands for 'Big Seven', 'MN' for 'Middle Nine', 'PAT' for 'Pan-Arctic Thousands', and PAS for 'Pan-Arctic Small watersheds. Note that the x-axis has a logarithmic scale. The color of the bars represents the mean bin catchment slope of the.**





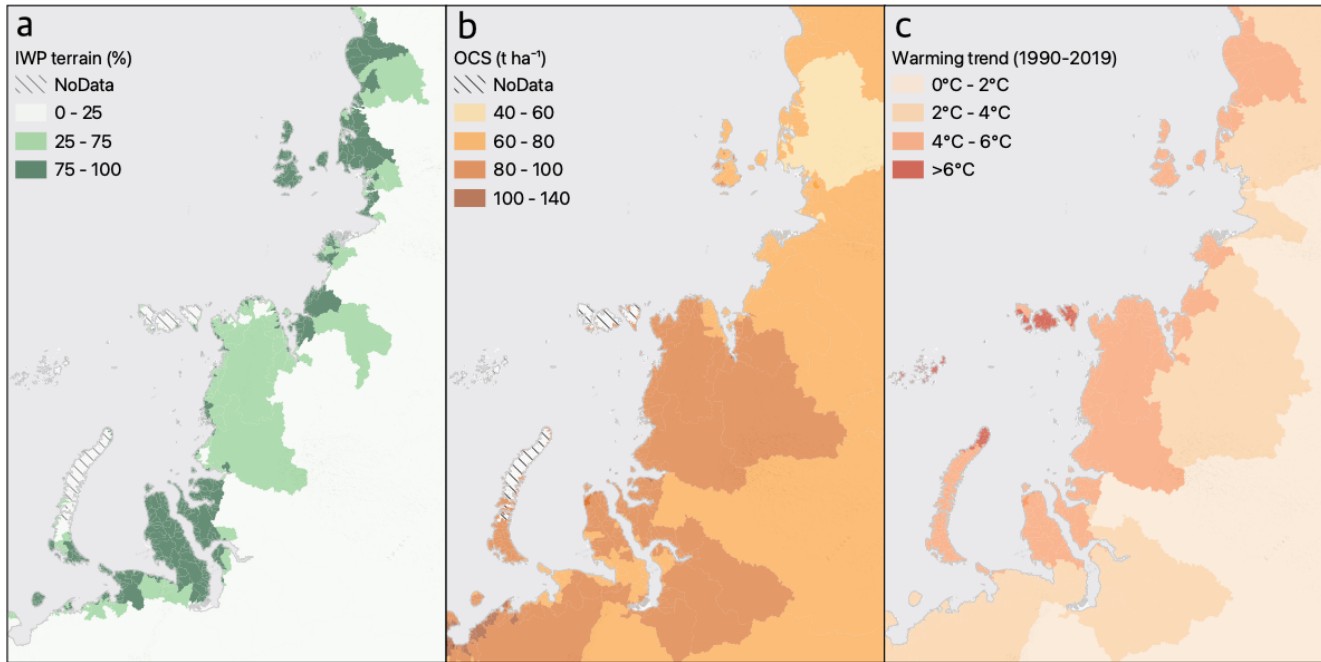

Figure 3. Siberian coastal watersheds with Ice wedge polygon (IWP) terrain (% watershed coverage) (a), soil organic carbon stock (OCS) in metric tons per hectare (b), and the mean watershed temperature trend taken over the period 1990 – 2019 (c) (map source: ARCADE database (Speetjens et al., 2022)).





**Figure 4. Correlations of binned data of selected catchment properties from our database. We calculated Spearman's Rho on the binned data. Most notably, we observe that small watersheds have experienced the greatest warming, while having the highest mean carbon stocks and the highest fraction of IWP terrain. Similarly, the data show that high OC stocks are found where most warming has occurred.**