# Peer review of "The Pan-Arctic Catchment Database (ARCADE)"

_Earth System Science Data, 2022_

## Author Response (AR1)

Dear Editor,

We thank the two reviewers for their constructive and positive comments.

We provide a point-to-point response (**in bold**) to the reviewer's comments (*in italics*) that addresses all raised issues.

On behalf of all authors,

Niek Jesse Speetjens

**Responses to reviewer comments**:
Reviewer 1:
*1) short chapter on soil properties (starting line 173): Please explain shortly for which purposes you calculated SOC contents (actually it should be clear, but please name it explicitly), but especially explain how you calculated the data, particularly for the individual depth intervals. How is the calculation based on SoilGrids, and how reliable is the data / how big do you think the uncertainties are, especially for the deeper soils?*

**We thank reviewer 1 for this suggestion and we agree that several sentences in this paragraph are unclear. Especially the part about depth intervals, as these were not calculated by us but are part of the SoilGrids data. We edited the text in the manuscript to clarify how SoilGrids data is included into the database. The revised paragraph now reads as follows:**

**"SoilGrids is a globally consistent dataset that contains soil properties (soil organic carbon (SOC) content (dg kg$^{-1}$), organic carbon density (dg dm$^{-3}$), nitrogen content (cg kg$^{-1}$), coarse fragments volumetric content (per 10000), sand, silt and clay content (g kg$^{-1}$), soil bulk density (cg cm$^{-3}$) for six depth intervals (0-5 cm, 5-15 cm, 15-30 cm, 40-60 cm, 60-100 cm, 100-200cm), and organic carbon stock (OCS) (t ha$^{-1}$) for the upper 30 cm of the soil) and classes (the most the likely soil class according to the World Reference Base (WRB) classification system (IUSS Working Group WRB, 2014)) at 250-meter resolution (Poggio et al., 2021). The ARCADE database aggregates soil property data from SoilGrids into watershed minimum, maximum, mean and standard deviation. OCS was also summarized into total watershed OCS (Gt) in the upper 30 cm of the soil. Soil class data from SoilGrids were summarized by calculating the fractional coverage of each class for each watershed. All watershed statistics were calculated using the "image.reduceRegion()" function in Google Earth Engine (Gorelick et al., 2017). We note that estimates of soil properties, especially for deeper soils, are often uncertain due to data scarcity in the permafrost region. We refer to Poggio et al., 2021 for more detailed discussions of uncertainties in the soil property projections."**

*2) Vegetation index (starting line 222): please explain shortly why you determined the vegetation indices only for the month of August*

**We thank reviewer 1 for this question. For the first version of the database we wanted to refrain from adding all possible columns (e.g. all months for all temporal data) since**

this would yield a database that becomes more complex to use. We considered August a representative month for the 'seasonal maximum NDVI' as vegetation biomass accumulates during the northern hemisphere summer. However, we realize that the annual maximum and 30-year trend hereof may serve as a better indicator. For this reason we recalculated the NDVI and NDVI-slope based on the annual maxima instead of only August.

*3) Table 4: what exactly is "grassland"? I am surprised that this land cover occurs in all catchment categories. Is there an explanation for this? I think that these are likely to be very different subcategories, i.e. "grassland" is not the same as "grassland". In the south of the large catchments (BS), "grassland" might already refer to parts of the extensive steppes / prairies. A similar question naturally arises for the categories "trees" and "shrubland". Unfortunately, this seems to me to be too rough or not clear enough. For hydrological modelling, for example, the categories are not explicit enough.*

We use the existing classification of landcover based on Zanaga et al. (2021). They classify grassland as "any geographic area dominated by natural herbaceous plants (plants without persistent stem or shoots aboveground and lacking definite firm structure): (grasslands, prairies, steppes, savannahs, pastures) with a cover of 10% or more, irrespective of different human and/or animal activities, such as: grazing, selective fire management etc. Woody plants (trees and/or shrubs) can be present assuming their cover is less than 10%. It may also contain uncultivated cropland areas (without harvest/ bare soil period) in the reference year." They also provide definitions for other categories (e.g., trees, shrubland, etc.). We agree that this classification may not provide the necessary nuance for all possible studies, and that the classification is not tailored specifically for the Arctic. However, we consider it a meaningful standardized classification. However, the database could be expanded into the future with additional classifications.

*4) Table 5: I assume the climatic variables derived from ERA5 have not yet been cross-checked for the catchments, have they? I would be interested to know, for example, to what extent the information on evaporation is correct (but that could be an investigation that follows the publication of the data).*

We thank reviewer 1 for this question. ERA5 reanalysis combines large amounts of historical observations using modelling and data assimilation. We did not back track the accuracy of the resulting product with ground observations. We agree that it could be compared into the future but consider that outside the scope of this study (as, for example, the suggested comparison with evaporation is an extensive, complicated, study by itself).

*5) Figure 1: How precise can a subdivision of catchments be on the Greenland ice sheet?*

Greenland icesheet catchments indeed come with large uncertainties. We have emphasized this better, in lines 137-139. Catchments have been delineated using the surface/ice-topography.

*6) When working through the manuscript, I felt that there were a few redundancies. I can't name them specifically, but I would like to ask the authors to check everything again for redundancies.*

**We appreciate this comment and have removed redundancies from the manuscript.**

*7) Lines 149-152: The example with snowmelt (0.021 m d-1): Sorry, but I didn't get it. Can you explain it differently?*

**Apologies for the unclarity. The explanation is as follows: We have included snow statistics such as the 30-year average value of the annual maxima of snow depth, snow cover, snow fall and snowmelt. These values hereby represent the 30-year average maximum monthly values. In the case of snowmelt, this is an indicator of the intensity of snowmelt during the melting season. We clarified this in the text. In future releases we plan to include more snow metrics.**

*8) Line 238: Something is missing here*

**Thank you. We corrected this to: …We "included" the tasseled cap indices...**

*9) Table 3: please add OCS to the legend / the explanations*

**We inserted the explanation in the caption.**

Reviewer 2:

*Table 2 and others. It is not clear in the description what the "n" stands for and it is not necessary at every title of the tables. Not only climatological but also physical parameters are provided. Word amount/sum/quantity in the column could replace "n". Consider if it is necessary to have this column in every table.*

**Thank you very much for your comments and suggestions. We include this column in every table because the number of catchments covered by the variables/parameters in the table varies between tables, because some of the used datasets did not cover the entire extent of the catchment delineations. We will change titles to exclude this explanation of n, change n into 'count' and explain the meaning in the caption of each table.**

*Table 3. It has to be edited as line number 560 was imprinted into the table.*

**We correct this in the revised manuscript.**

*Provided link took me to "ARCADE: The pan-ARctic CAtchment DAtabase" – it should be DatabasE.*

**Thank you. We have corrected this error.**

*Line 79. understanding **the** impacts of climate*

**Edited in text.**

*Line 161. time series*

**Edited in text.**

*Line 162. follows by*

**Edited in text.**

*Line 191. the occurrence*

**Edited in text.**

*Line 200. word converted is used twice in this sentence*

**Edited in text.**

*Line 219. the presence and the occurrence*

**Edited in text.**

*Line 307. Remove one of two dots.*

**Edited in text.**

---

## Author Response (AR2)

[revised manuscript text omitted]

ARCADE Watersheds

'The Big 7'

1. Ob
2. Yenisey
3. Lena
4. Mackenzie
5. Nelson
6. Yukon
7. Kolyma

'The Middle 9'

8. Severnaya Dvina
9. Indigirka
10. Pechora
11. Olenek
12. Thelon
13. Yana
14. Kathanga
15. Pyasina
16. Taz

**Figure 1. Circumpolar map of all ARCADE watersheds, 1 km$^2$ and larger, Strahler order 5 and higher, at 90-meter resolution with insets of the Southern Beaufort Sea region (upper left) and the Laptev Sea coast, including the New Siberian Islands (upper right). (Background map: International Bathymetric Chart of the Arctic Ocean V4.0 (IBCAO) (Jakobsson et al., 2020)).**

615

[Figure]

**Figure 2. The distribution of watershed areas in the pan-Arctic watersheds database and the range of the four groups that classify based on watershed area. 'BS' stands for 'Big Seven', 'MN' for 'Middle Nine', 'PAT' for 'Pan-Arctic Thousands', and PAS for 'Pan-Arctic Small watersheds. Note that the x-axis has a logarithmic scale. The color of the bars represents the mean bin catchment slope of the.**

625

630

[Figure]

**Figure 3. Siberian coastal watersheds with Ice wedge polygon (IWP) terrain (% watershed coverage) (a), soil organic carbon stock (OCS) in metric tons per hectare (b), and the mean watershed temperature trend taken over the period 1990 – 2019 (c) (map source: ARCADE database (Speetjens et al., 2022)).**

[Figure]

635

**Figure 4. Correlations of binned data of selected catchment properties from our database. We calculated Spearman's Rho on the binned data. Most notably, we observe that small watersheds have experienced the greatest warming, while having the highest mean carbon stocks and the highest fraction of IWP terrain. Similarly, the data show that high OC stocks are found where most warming has occurred.**

640